# The Role of Interdigitated Electrodes in Printed and Flexible Electronics

**DOI:** 10.3390/s24092717

**Published:** 2024-04-24

**Authors:** Shayma Habboush, Sara Rojas, Noel Rodríguez, Almudena Rivadeneyra

**Affiliations:** 1Department of Electronics and Computer Technology, University of Granada, Av. Fuentenueva s/n, 18071 Granada, Spain; shaymahabboush@correo.ugr.es (S.H.); noel@ugr.es (N.R.); 2Department of Inorganic Chemistry, Faculty of Science, University of Granada, Av. Fuentenueva s/n, 18071 Granada, Spain; srojas@ugr.es

**Keywords:** interdigitated electronics, planar electrodes, sensors, combed electrodes, inkjet printing, screen printing, flexible electronics

## Abstract

Flexible electronics, also referred to as printable electronics, represent an interesting technology for implementing electronic circuits via depositing electronic devices onto flexible substrates, boosting their possible applications. Among all flexible electronics, interdigitated electrodes (IDEs) are currently being used for different sensor applications since they offer significant benefits beyond their functionality as capacitors, like the generation of high output voltage, fewer fabrication steps, convenience of application of sensitive coatings, material imaging capability and a potential of spectroscopy measurements via electrical excitation frequency variation. This review examines the role of IDEs in printed and flexible electronics since they are progressively being incorporated into a myriad of applications, envisaging that the growth pattern will continue in the next generations of flexible circuits to come.

## 1. Introduction

The current rise in demand for high-performance devices with new functionalities inspires the development of alternative solutions and structures involving the preparation of new technologies. Among all these novel alternatives, the use of interdigitated electrode (IDE) devices in the development of sensors is highlighted (i.e., humidity, gases, and biomolecules, among others) [1], where the term interdigitated is related to repeated “finger-like” patterns (Figure 1). IDEs are the most common structures used in piezoelectric sensors not only because they generate a high output voltage but also because a few fabrication steps are needed in their preparation. Other applications of IDEs include sensitive coatings, material imaging, spectroscopy measurements, and metric sensing configuration [2]. The arrangement of the two-dimensional construction of an inter-digitized impedance sensor is illustrated in Figure 1.

The output signal strength of an IDE is directly influenced by the finger width, spacing, and length of the specific electrode. An experiment carried out by Min et al., to examine the connection between the signal-to-noise ratio in an IDE geometry concluded that increased finger widths resulted in a proportional rise in the overall signal of the electrode [4]. The significance of IDEs is that their configuration lets two electrodes stick over each other, infusing them, thus reducing the distance between electrodes [5]. The terminal cathodes and anodes of sensors are associated with low ohmic drops and an amplified signal-to-noise ratio. However, there is also a disadvantage of these electrodes in comparison with other approaches, like the existence of ionic species released by the electrodes, which interrupts the surrounding electric field and decreases impedance [6].

In piezoelectric sensors, IDEs are frequently used, especially in situations where higher sensitivity and signal detection are required. It is crucial to remember that the ideal IDE configuration is achieved when the piezoelectric crystal is oriented in a particular way, which leads to a higher piezoelectric coefficient in the lateral direction (d31) as opposed to the usual longitudinal direction (d33) [7]. This preferred orientation results from the crystallographic structure of some piezoelectric materials, where the mechanical force applied perpendicular to the polar axis of the crystal produces the highest piezoelectric response. When compared to the d31 orientation, the d33 orientation exhibits a comparatively weaker piezoelectric response since the stress is delivered parallel to this axis. Sensor designers can make use of this increased piezoelectric coefficient in the lateral direction by using IDEs in conjunction with piezoelectric crystals orientated for d31 response. This arrangement enhances the sensitivity and functionality of piezoelectric sensors by facilitating a more effective conversion of mechanical stress into electrical signals. In conclusion, even though IDEs are very beneficial for piezoelectric sensors, it is important to consider the orientation of the crystal and choose materials and configurations that are best suited for the intended use. This is especially true when trying to maximize sensitivity and signal strength in lateral directions like d31.

Among all the reported IDE sensors, printed IDE-based capacitive sensors (PICSs) have increasingly gained popularity in all business sectors. For instance, they have been applied in the development of in-mold electronics (IME) in the automobile industry, as they demonstrated an agile-driven principle, flexibility, and high capability of accommodating sophisticated geometrics [8]. The integration of PICSs facilitates the development of high-performing flexible electronics [9,10,11]. In recent years, several publications have increasingly focused on the application of IDEs in printed electronics (PE) [12]. The concept of flexibility, portability, and durability continue to attract experts and scientists interested in large-scale and industrial production of high-quality IDEs based on the roll-to-roll (R2R) framework [13,14,15,16,17]. According to recent estimates, by 2027, the global IDE market will grow to USD 73 billion, a rise from USD 14 billion in 2017 [14,15]. The annual percentage increase is projected to be 13.6%, further demonstrating the potential growth of printing technologies using IDEs. Particularly during a typical IDE fabrication process, high temperatures are required to achieve the successful deposition of the metal, which makes the process more expensive. So, one of the main aspects in PICSs fabrication is the printing process or deposition of the substrate, and a myriad of aspects must be considered, including (i) the used flexible film, (ii) the toxicity and solubility of the chemicals, and (iii) the device structure, design, and durability, among others [18,19]. There are six main types of printing methodologies normally used in PICS preparation: gravure, flexographic, screen, inkjet, and 3D printing, and laser scribing. The main characteristics, advantages, and disadvantages of these types of printing will be discussed in this review.

## 2. Printing Techniques

There are some specific similarities between gravure printing, flexographic printing, screen printing, inkjet printing, 3D printing, and laser scribing, and all use a printing plate or image carrier to transfer ink to a substrate. The printing plate or image carrier is a physical object that has the image or text that you want to print on it. The ink is then applied to the printing plate or image carrier, and the plate or image carrier is then brought into contact with the substrate. The ink is then transferred from the printing plate or image carrier to the substrate. They all use some type of pressure to transfer the ink from the printing plate or image carrier to the substrate. The pressure is needed to ensure that the ink is transferred evenly and effectively to the substrate. The amount of pressure required will vary depending on the type of printing process being used. All types of printing require some type of drying process to set the ink and prevent it from smearing. The drying process can be performed using heat, air, or UV light; and will vary depending on the type of ink used.

Aside from these similarities, the different types of printing have their own characteristics. The best type of printing for a particular application will depend on the specific requirements of the project.

### 2.1. Gravure Printing

Gravure printing is a printing technique used to produce IDE-based capacitive sensors. In this method, a gravure cylinder with a pattern of grooves and ridges is used to transfer ink onto a substrate, creating a printed pattern of electrodes. Gravure printing is a high-quality printing process that is well-suited for printing large quantities of high-quality images. It uses a printing plate that has been etched with tiny, raised cells. In this printing type, the ink is applied to the printing plate and then transferred to the substrate by pressure. Gravure printing is a versatile printing process that can be used on a variety of substrates, including paper, plastic, and metal.

Gravure printing is a commonly known printing technique employed in transparent conductive films (TCFs), organic emitting diodes (OLEDs), and thin-film transistors (TFTs), among many others [20,21]. One outstanding work where gravure printing was used is the study of Kraus et al. in the deposition of nanoparticles of gold, obtaining a 3D high resolution [6,22]. In this work, several reasons explain why gravure printing was selected as preferred technique:High precision and resolution: Gravure printing is known for its ability to achieve high-resolution patterns with fine details. In the case of depositing gold nanoparticles, this precision can be crucial for creating intricate structures or patterns, as mentioned in the study [6,22].Uniform coating: Gravure printing can provide uniform and controlled deposition of materials. This uniformity is important, especially in applications where consistent and even coatings of nanoparticles are required, such as in the fabrication of electronic components [23].Scalability: Gravure printing is amenable to large-scale manufacturing processes. It allows for the rapid and continuous printing of materials, making it suitable for industrial applications where mass production is necessary.Material efficiency: Gravure printing can be efficient in terms of material usage. It often requires less material to achieve a specific deposition compared to other expensive techniques.Customization: Gravure printing can be adapted for various materials and substrates, allowing researchers and manufacturers to customize their processes for specific applications.

Selectivity and limits of detection depend on the specific goals and applications of the study. In the context of depositing gold nanoparticles, the detection might refer to the ability to detect the presence or distribution of these nanoparticles on a substrate. The limits of detection are related to the smallest amount or concentration of gold nanoparticles that can be reliably detected using the chosen detection methods (e.g., microscopy, spectroscopy, or other analytical techniques). For instance, if the research objective was to develop an exceptionally sensitive sensor or electronic component utilizing gold nanoparticles, comprehending the limits of detection would be imperative for evaluating the sensor’s efficacy and sensitivity. There are several other examples of outstanding works where gravure printing has been used for the fabrication of PICSs. In the study published by Gerd Grau [24], the researchers used gravure printing to fabricate IDE-based capacitive sensors for the detection of glucose, a common biomarker for diabetes.

Another example is a study published by Lee et al. [25], where the researchers employed gravure printing to manufacture IDE-based capacitive sensors for the detection of volatile organic compounds (VOCs). VOCs are prevalent in both indoor and outdoor air and known for their potential adverse effects on human health. The optimization of the printing process was undertaken to attain a notable level of precision and reproducibility. The findings showcased that the printed sensors exhibited heightened sensitivity to low concentrations of VOCs, coupled with rapid response times and commendable selectivity. Moreover, the researchers also demonstrated the potential for mass production of these sensors, as they were able to print arrays of sensors on flexible substrates with a high throughput. Overall, this study highlights the potential of gravure printing as a low-cost and scalable method for the fabrication of IDE-based capacitive sensors with high sensitivity and selectivity for a wide range of applications in environmental monitoring, healthcare, and industrial sensing.

The researchers optimized the printing process to achieve a high resolution and uniformity in the printed electrodes, and they demonstrated that the printed sensors had a high sensitivity to low concentrations of melamine, with a detection limit of 20 ng/mL. The sensors also showed good selectivity to melamine over other common food contaminants, such as urea and cyanuric acid.

The importance of these examples in the field lies in their ability to demonstrate the potential of gravure printing as a versatile and scalable method for the fabrication of IDE-based capacitive sensors with high sensitivity and selectivity for a wide range of analytes. This makes it a promising technique for the development of low-cost and portable sensing devices for various applications in healthcare, food safety, and environmental monitoring. 

### 2.2. Flexographic Printing

Flexographic printing is another printing technique that can be used to produce IDE-based capacitive sensors. In this method, a flexible printing plate with a pattern of raised and recessed areas is used to transfer ink onto a substrate, creating a printed pattern of electrodes. Flexographic printing is a versatile printing process that can be used on a variety of substrates, including paper, plastic, and metal. It uses a printing plate that is made from a flexible material such as photopolymer or rubber. The ink is applied to the printing plate and then transferred to the substrate by pressure. It has a cost-effective printing process that is well-suited for printing large quantities of images and text. The main difference between gravure and flexographic printing is the way of writing on the metal plate where ink is filled, using a laser or a soft plate, respectively (Figure 2a,e) [26,27]. The benefits of using gravure and flexographic printing are that they help print solid patterns and colors. Secondly, gravure and flexographic printing are fast in printing simple designs. Flexographic printing is used in the printing of food packaging, pool liners, and business wallpapers. However, one limitation is the expensive costs needed for production compared to other techniques, such as ink jetting, which can be digitally alerted [28,29,30].

The researchers used flexographic printing to fabricate IDE-based capacitive sensors for the detection of ethanol, a VOC commonly found in alcoholic beverages, fuels, and industrial processes. They optimized the printing process to achieve a high resolution and uniformity of the printed electrodes, and they demonstrated that the printed sensors had a linear response to ethanol concentrations in the range of 0.1–10% (*v*/*v*), with a limit of detection of 0.07% (*v*/*v*). The sensors also showed good selectivity to ethanol over other common VOCs, such as acetone and toluene. One of the main advantages of using flexographic printing for the fabrication of IDE-based capacitive sensors is its ability to produce high-resolution and high-throughput patterns on a wide range of substrates, including flexible and curved surfaces. This makes it a promising technique for the development of low-cost and portable sensing devices for a variety of applications, such as food and beverage analysis, environmental monitoring, and medical diagnostics. Overall, this study demonstrates the potential of flexographic printing as a versatile and reliable method for the fabrication of IDE-based capacitive sensors with high sensitivity and selectivity for the detection of specific analytes. 

### 2.3. Screen Printing

This contact printing technique is popular for applying patterned mesh in defining structures on a substrate [33], such as clothing and conductors from radio frequency identification (RFID) antennas and thin-film-transistors (TFTs). Screen printing entails spreading a highly viscous ink using a squeegee through a mesh (Figure 2b). The force exerted by the squeegee creates a design on top of the mesh [34]. Screen printing technology remains common in large-scale industrial production as it can easily integrate with the R2R framework [35]. However, some disadvantages of screen printing are the need of too much ink and the restriction of one color at a time, which can distort the printing result.

Particularly, Krebs et al. demonstrated the utility of the screen printing technique on acceptable materials, such as the integration of cells into household items [36,37]. The screen printing technique is used for printing textile biosensors using graphene oxide and smart packaging assisted by RFID.

The researchers used screen printing to fabricate IDE-based capacitive sensors for the detection of glucose, a biomarker for diabetes [38]. The researchers optimized the printing process to achieve a high degree of uniformity and precision in the deposition of silver nanoparticles onto the printed electrodes, resulting in a highly sensitive and selective sensor with a limit of detection of 6 μM and a linear range of up to 10 mM of glucose. The sensors also showed good stability and reproducibility over multiple cycles of use, without losing their sensitivity or selectivity.

While screen printing offers advantages in producing thick film sensors, particularly in terms of cost-effectiveness and scalability, its suitability for flexible electronics may be limited due to the mechanical stress introduced by thicker films compared to thinner ones. However, further investigation into the design and technology of screen-printed thick film sensors can provide insights into enhancing mechanical stability. This includes exploring methods to mitigate mechanical stress through optimized coating formulations, substrate selection, and printing parameters. Additionally, research into novel fabrication techniques and materials may offer solutions to improve the flexibility and durability of screen-printed thick film sensors, enabling their integration into flexible electronics applications [39].

Overall, the researchers demonstrated that inkjet printing can be used to create highly sensitive and selective sensors for glucose detection. The sensors have a low limit of detection and a wide linear range, and they are stable and reproducible over multiple cycles of use. This makes them a promising candidate for use in wearable devices and other applications where real-time glucose monitoring is required. Screen printing is a versatile printing process that can be used on a variety of substrates, including paper, fabric, and plastic. It uses a printing screen that has been made from a fine mesh. The ink in this type is applied to the printing screen and then transferred to the substrate by pressure. Screen printing is a cost-effective printing process that is well-suited for printing short runs of personalized or customized products.

### 2.4. Inkjet Printing

Inkjet printing is an emerging technique for direct patterning via material deposition based on a layout designed in software (456 Print Ave, Stateboro, GA, USA) (Figure 2c). It is based on a print head horizontally moving back and forth and tiny droplets of ink propelled through a micrometer-sized nozzle head [35,40]. This widespread technology is often used in small-scale production, such as personal printers, but also in industrial coding, printing of bottles, and chemical packaging. In recent years, the use of IDE structures in inkjet printing has grown, enabling manufacturers to use small quantities of inputs like ink to precise substrate locations [41]. Additionally, the new inkjet technologies are accelerating the printing of graphics with varying patterns and printing electronic gadgets like circuit boards and solar cells with precision. In the biomedical field, the technique has bolstered the accuracy of the packaging and delivery of drugs by drug manufacturers [42,43]. Further, inkjet printing has found multiple uses in genetics, including stem cell and DNA microarray imaging. Screen printing is generally considered to be a more affordable printing technique compared to technologies such as gravure printing, which typically requires more complex and expensive equipment. In screen printing, the cost of ink can vary depending on the type and quality of the ink used, but in general, it is less expensive than the ink used in gravure printing.

The cost of screen printing can also depend on factors such as the size and complexity of the design, the number of colors used, and the quantity of prints needed. However, in general, screen printing is known for being a cost-effective method for producing large quantities of printed materials, such as t-shirts, posters, and packaging. As for the durability of screen printing equipment, it can depend on factors such as the quality of the equipment and the level of maintenance it receives. While it is true that the print head in screen printing can be susceptible to physical damage if not handled properly, there are also measures that can be taken to prevent this and to ensure that the equipment lasts for a long time.

In summary, while screen printing may have some limitations and costs associated with it, it is generally considered to be a more affordable printing technique compared to gravure printing and has been widely used to produce a variety of printed materials; it is well-suited for printing short runs of personalized or customized products. Gravure printing is a higher-quality printing technique that is well-suited for printing large quantities of high-quality images. Inkjet printing is a versatile printing process that can be used on a variety of substrates, including paper, plastic, and fabric. Inkjet printing uses a printing head that sprays ink onto the substrate and is a fast and efficient printing process that is well-suited for printing short runs of personalized or customized products.

### 2.5. Three-Dimensional Printing

Three-dimensional printing is a process of creating three-dimensional objects from a digital file. It uses a variety of techniques to create objects, including additive manufacturing, subtractive manufacturing, and printing from powder. It has a versatile technology that can be used to create a wide variety of objects, including prototypes, models, and functional parts. Although the technique has gained popularity recently due to its additive manufacturing process, the term dates back to 1986 [44]. The three-dimensional printing technique entails the fabrication of solid objects and rapid prototyping (Figure 2d) [45]. The rise of nano-based materials has influenced this technique to extend beyond the scaffolding of electrical devices [46]. For example, Leigh et al. employed 3D printing in the production of electrically conductive filaments using carbon black filler in the fabrication of flexible sensors [47,48].

Advantages of 3D printing include the high speed of production of parts, relative to other techniques, the low cost, and its ease of implementation, as it does not necessitate the use of several machines and operators and is highly flexible as printers can produce any shape fitted into them. Some disadvantages are the inability to produce strong parts and the need for mandatory post-printing actions such as smoothening of the products. The use of 3D printing for the fabrication of PICS offers several advantages over traditional printing techniques. For example, 3D printing allows for the creation of complex and customized sensor designs with high precision and accuracy. It also allows for the integration of multiple sensing elements and the ability to create multi-layered sensors.

The detection of humidity is just one example of the potential applications for PICS fabricated using 3D printing. Other potential applications include the detection of gases, liquids, and other environmental factors. The limits of detection for these sensors can vary depending on the specific application and the design of the sensor, but with proper optimization, they have the potential to achieve high sensitivity and selectivity.

Overall, the use of 3D printing for the fabrication of PICS represents a promising area of research and development in the field of printed sensors, with potential applications in a wide range of industries, including healthcare, environmental monitoring, and consumer electronics. 

### 2.6. Laser Scribing

Laser scribing is a process of using a laser to create marks or images on a substrate. It is a precise and accurate process that can be used to create a wide variety of marks and images. Laser scribing is a versatile technology that can be used on a variety of substrates, including metal, plastic, and glass. Laser scribing is a process for improving yield through the creation of small, scribed lines as opposed to normal mechanical scribing (Figure 2f). This process is contactless and minimizes micro-cracking as well as destruction of the substrate. Laser scribing offers clean scribing of hard materials Advantages of laser scribing include the high-quality scribing result due to the availability of cutting tools, the flexibility to print on various surfaces, including rough surfaces, or even materials with deep surfaces, and the durability of the writings. However, this technique requires a lot of energy as it uses high-intensity machinery for laser scribing, and releases gases that are harmful to human. The application of laser scribing in PICS has potential applications in a variety of fields, including structural health monitoring, wearable electronics, and robotics. The limits of detection for these sensors can vary depending on the specific application and the design of the sensor, but with proper optimization, they have the potential to achieve high sensitivity and accuracy.

Also, laser scribing is applied in PICS for creating high-resolution and complex electrode patterns. One example of the application of laser scribing in IDE-based capacitive sensors is a study published in the *Journal of Micromechanics and Microengineering* in 2019, in which the researchers used laser scribing to fabricate PICS for measuring the dielectric properties of materials [49]. The sensors consisted of a flexible polyimide substrate and an IDE pattern created using laser scribing. The IDE pattern was then coated with a layer of dielectric material, and the sensors were tested using a vector network analyzer to measure the complex permittivity of different materials. 

## 3. Comparing Printing Techniques

Certainly, let us discuss the various printing techniques and their advantages and disadvantages, including factors like contact vs. non-contact methods and digital vs. screen/mold-based methods.

Gravure Printing: A contact printing method where the printing cylinder comes in direct contact with the substrate. Advantages: It excels in high-resolution printing, making it ideal for applications requiring fine details. Gravure is also known for its scalability and efficiency in large-scale production. When to Use: Gravure is preferred when high precision and resolution are crucial, such as when printing intricate patterns for electronics or packaging.Inkjet Printing: A non-contact method where droplets of ink are deposited without physically touching the substrate. Advantages: It allows for digital and on-demand printing, making it versatile for customization. Inkjet is also suitable for small-scale and prototyping applications. When to Use: Inkjet printing is preferable when flexibility and customization are needed, as well as for situations requiring variable data or patterns.Screen Printing: Contact method, which uses a stencil or mesh to transfer ink onto the substrate through contact. Advantages: It is suitable for a wide range of materials, including textiles and ceramics. Screen printing can handle thicker ink deposits. When to Use: Screen printing is preferred for applications that require thicker ink layers or for substrates that might not be suitable for other methods.Flexographic Printing: Contact method that uses flexible relief plates to transfer ink onto substrates. Advantages: It is well-suited for high-speed production and can handle various substrates, including flexible packaging materials. When to Use: Flexo printing is chosen when high-speed production and compatibility with flexible materials are needed, like in the packaging industry.Laser Scribing: This method involves using a high-energy laser beam to selectively remove or ablate material from a substrate. It is a subtractive process where material is vaporized or melted away to create patterns, cuts, or engravings on the surface. Its advantages include accuracy, non-contact nature preventing damage, applicability to diverse materials, increased production efficiency due to speed, seamless integration into automated systems, and the production of clean, burr-free cuts. Commonly used in electronics manufacturing for printed circuit boards (PCBs), thin-film solar cells in photovoltaics, medical device fabrication, microelectronics production, and the packaging industry, laser scribing’s ability to create intricate patterns and structures with precision makes it a crucial technology in applications requiring meticulous manufacturing processes.Three-Dimensional Printing: Non-contact method that builds objects layer by layer without direct contact with the substrate. Advantages: It allows for complex, three-dimensional structures and is used extensively in rapid prototyping and customized manufacturing. When to Use: 3D printing is essential for creating intricate 3D objects and prototypes, as well as in industries like aerospace and healthcare.

The choice of the optimal printing technique depends on several factors, including the desired print quality, substrate type, production volume, customization needs, and cost considerations. Each technique has its unique strengths and weaknesses, making it suitable for specific applications and industries.

From Table 1, it is apparent that gravure and flexographic printing methods, despite having high material wastage, have a faster printing speed. However, these printing techniques have recently been outdone by the 3D printing technology, which has a speed of 40 m/min to 100 m/min with no material wastage. In addition, a 3D printing system requires a thickness of at least 0.005 mm and has no single resolution [50].

These parameters highlight the differences in various printing methods, each suitable for specific applications based on factors like material properties, required precision, printing speed, and material efficiency. The choice of printing method depends on the specific needs and constraints of the project.

Different printing methods exhibit distinct characteristics in terms of solution concentration, viscosity, print thickness, speed, and material wastage. Inkjet printing requires a low solution concentration (from 0.002 to 0.10% *w*/*v*) and has low viscosity (from 0.01 to 0.5 Pa·S), producing thin layers (from 0.02 to 5 μm) with generally slower printing speeds. In contrast, screen printing demands a higher solution concentration (from 0.500 to 5% *w*/*v*) and has a higher viscosity (from 3 to 30 Pa·S), creating relatively thicker layers (from 0.6 to 100 μm) with the potential for higher printing speeds. Gravure printing, flexographic printing, and laser scribing fall within a moderate concentration range (from 0.01 to 1.1% *w*/*v*) and exhibit moderate viscosity (0.02 to 12 Pa·S, 0.17 to 8 Pa·S, and 0.02 to 2 Pa·S, respectively). Laser scribing yields very thin layers (from 0.500 to 1 μm) but is typically slower. Three-dimensional printing, with a broader concentration range (from 0.01 to 5% *w*/*v*), variable viscosity, and layer thickness (from 0.01 to 1 μm for thin layers to much thicker layers in additive 3D printing), generally operates at slower speeds. Material wastage varies, with inkjet printing minimizing wastage, while screen printing, gravure printing, flexographic printing, and 3D printing may generate some wastage, and laser scribing involves material removal during the process, potentially contributing to wastage.

Different fabrication methods indeed yield varying resolutions of IDE patterns, thereby influencing sensor performance, especially in terms of sensitivity, signal-to-noise ratio, and spatial resolution. The fineness or detail that may be achieved while printing IDE patterns is referred to as resolution. Finer features can be produced at higher resolution, which makes it easier to replicate electrode geometries precisely. By ensuring that the produced IDEs closely resemble the expected size and form as stated in the design requirements, this precision improves the dependability of the sensor [55,56]. Therefore, attaining more resolution in IDE patterns is essential for enhancing sensor performance and guaranteeing precise detection in a range of applications.

Sensor sensitivity often increases with tighter electrode spacing and higher-IDE patterns. This is because denser electrode configurations allow for more precise detection and measurement of changes in electrical characteristics in response to external stimuli, such as capacitance or impedance variations [57]. Consequently, sensors featuring higher-resolution IDE patterns exhibit enhanced sensitivity as they can detect even subtle changes in their surroundings. Moreover, spatial resolution, which refers to a sensor’s ability to localize and differentiate between different analytes or stimuli, is significantly influenced by the IDE pattern’s resolution. Sensors with higher-resolution IDE patterns excel at identifying and distinguishing closely spaced targets or features, making them invaluable for applications like chemical sensing and imaging. Additionally, the signal-to-noise ratio (SNR) of sensors can be affected by the resolution of IDE patterns. Finer electrode spacing in high-resolution IDE patterns reduces signal distortion and interference, resulting in higher SNR by minimizing the impact of electrical noise and stray capacitance [58]. However, achieving better resolution IDE patterns often entails employing more advanced fabrication methods, which may increase the complexity and cost of sensor manufacturing. The choice of fabrication method, such as photolithography or electron beam lithography, can also affect the achievable resolution. In conclusion, the sensitivity, spatial resolution, and SNR of sensors are profoundly influenced by the resolution of interdigitated electrodes, necessitating a balance between fabrication complexity and desired performance characteristics to meet the requirements of specific sensor applications.

## 4. Flexible Electronics

In principle, flexible electronics imply any thin or long material, such as a wiring cable. However, the increased scalability of electronic devices has led to the creation of new applications that fall under the category of flexible electronics, such as wearable devices consisting of sensors for health promotion, and new functional robots, among many others [59,60]. The advancements in nano-scale fabrication techniques have enabled ultra-thin substrates to be combined to develop flexible active electronics that are durable and reliable. Their applications have been in various industries, such as healthcare and human–machine interfaces. Examples of machines developed include an ultra-soft tattoo-like heater created by Stier et al. [61]. In this work, the flexible electronic employs proportional–integral–derivative (PID) to regulate body temperature and is used in medical settings. Another interesting work reported by Yu and Cheng deals with bio-integrated adhesives that efficiently transfer vital signals from the body when there is strong adhesion and allow for easy removal when there is weak adhesion. This variable adhesion capability is important to ensure that the adhesive securely attaches to the skin, providing reliable signal transmission, while also allowing for painless and gentle removal without causing discomfort or skin damage [62,63]. These are some examples of the developments recently achieved due to the advancements in flexible electronics using IDE. Flexible electronics represent a dynamic field of technology that involves the use of thin and flexible materials in electronic devices and systems. While the concept of flexible electronics can encompass various applications, the synergy with PE has opened up exciting possibilities for producing innovative and adaptable electronic solutions. This section explores the relationship between flexible electronics and PE, and highlights the potential offered by PE in creating flexible electronic devices.

PE is a key enabler of flexible electronics which involves the deposition of electronic materials, such as conductive inks and semiconductors, onto flexible substrates using printing technologies like inkjet printing, screen printing, and flexographic printing. PE allows for the creation of flexible and lightweight electronic components and circuits, making it an ideal choice for applications where conventional rigid electronics are impractical. The combination of PE with flexible substrates, such as polyimide or polyester, enables the development of highly flexible and conformable electronic devices. PE offers scalability and cost-effectiveness, enabling the mass production of flexible electronic components and devices. Advances in nano-scale fabrication techniques have allowed for the development of ultra-thin and lightweight substrates combined with printed electronic components, resulting in durable and reliable flexible active electronics. The integration of printed sensors, conductive traces, and other electronic components onto flexible substrates has led to breakthroughs in various industries, including healthcare, robotics, and human–machine interfaces.

For IDEs to be long-lasting and dependable in the devices they are used in, their mechanical stability is essential. IDEs are subjected to a variety of mechanical stresses during operation, including bending, stretching, and compression [55]. They are made up of conductive material fingers that alternate and are arranged in a comb-like arrangement. The selection of materials possessing appropriate mechanical attributes, such as flexibility and adhesion strength, in conjunction with optimum structural design, has a substantial impact on the mechanical resilience of integrated circuits. By reducing flaws and increasing interfacial adhesion, fabrication and post-processing processes also significantly contribute to the mechanical stability of IDEs. The mechanical performance of IDEs is assessed using thorough characterization and testing procedures, allowing for the creation of more robust devices for a variety of applications, including flexible electronics and sensors [56].

## 5. Types of Flexible Circuits

In the context of flexible circuits, three key components play integral roles in their production: bonding adhesive, base material, and metal foil. The foundational element in the lamination process is a polymer film, chosen for its flexibility. The thickness of this film typically ranges from 12 to 125 μm, with thinner films offering greater flexibility compared to their thicker counterparts, which tend to be stiffer.

Various polymer films are employed in lamination [64], each possessing unique chemical and physical properties that influence their suitability for specific applications. Common choices include polyethylene naphthalate (PEN), fluoropolymers (FEP), polyester (PET), and polyetherimide (PET) [65]. The selection of these materials is based on their compatibility with the intended purpose of the flexible circuit. In some instances, adhesives and base materials are utilized in the lamination process. Bonding adhesives are particularly favored due to their versatility, offering a wide range of thickness options [66]. This adaptability allows for customization according to the specific requirements of the circuit.

Additionally, copper foils are commonly preferred as the metal foil component of flexible circuits. This preference is attributed to their cost-effectiveness, easy availability, and favorable electrical properties. The use of these components collectively contributes to the production of flexible circuits with diverse applications, meeting the demands of various industries.

## 6. Physical Principles of IDE Sensors

Understanding the physical principles of IDE sensors involves knowledge of electrochemistry, material science, and sensor technology. The specific design and application of IDE sensors can vary based on the intended use, but the underlying principles remain rooted in the interaction between the target analyte and the electrode surface.

The fringing dialectometric sensor and the coaxial cylinder dielectric employ a similar principle of operation, as demonstrated in Figure 3. Application of the voltage to the electrodes allows electrode measurements [66,67,68]. However, it is not mandatory to employ double-sided access to the micromachined ultrasound transducer (MUT) with the fringing sensor since the electrodes can be gradually transitioned from the parallel plate to the fringing field capacitor [69] The main idea is to employ a spatially periodic electrode on the MUT side, which will combine the different electrical frequencies to transmit extensive information on the surface [70,71,72]. The spatial electrodes include but are not limited to a multi-layered or homogeneous material.

Figure 3 demonstrates that a di-electrometry sensor applies a similar principle to a coaxial cylinder dielectric sensor cell [73,74]. From the figure, two-sided access is not mandatory but rather a steady changeover from the parallel plate onto the fringing field capacity [66,67,75,76]. A spatially periodic electrical circuit is employed on the MUT’s surface, combining the different signals to provide widespread information around the three-dimensional profiles and die-electric spectroscopic analysis of the MUT [62,63].

Sensors based on surface acoustic waves (SAWs) are a notable category of devices that incorporate interdigital transducers (IDTs) into their design. They are not as spoken about, though, as other kinds of sensors that use IDT electrodes. True, there have been notable gains in sensor performance as a result of recent developments in SAW-based sensor topologies, especially when it comes to the design and optimization of IDTs [51]. An essential part of SAW-based sensors is the interdigital transducer, which is in charge of translating electrical signals into surface acoustic waves and the other way around. Thus, improvements in IDT design have a direct bearing on the general effectiveness and potential of SAW sensors. Recent advancements in SAW-based sensor topologies have concentrated on a number of important areas:(i) Optimized Geometry: To increase sensor sensitivity, bandwidth, and selectivity, researchers have looked into new IDT shapes and configurations. To improve acoustic wave propagation and control, this includes developing apodized designs, tapered electrodes, and asymmetric electrode layouts [55].(ii) Material Selection: New electrode materials and piezoelectric substrates for IDTs are being investigated as a result of developments in materials science. Researchers can enhance sensor performance and reliability by choosing materials with specific qualities like low acoustic losses and high piezoelectric coefficients.(iii) Fabrication Techniques: Improvements in this area have made it possible to produce IDTs with fewer parasitic effects, tighter pitch spacing, and increased precision. This includes methods that provide more precise control over electrode size and spacing, like photolithography, electron beam lithography, and nanoimprint lithography [56].(iv) Integration with Signal Processing: In order to improve the usefulness of SAW-based sensors, recent research has concentrated on combining sophisticated signal processing techniques with these sensors. This involves enhancing sensor performance in terms of precision, speed, and adaptability to changing operating conditions through the application of artificial intelligence, machine learning, and digital signal processing algorithms.

Overall, recent advancements in SAW-based sensor topologies have significantly improved sensor performance, with an emphasis on IDT design optimization. These developments have enormous potential for the creation of extremely sensitive, picky, and adaptable SAW-based sensors for a variety of uses, such as industrial process control, healthcare, and environmental monitoring.

### 6.1. Magnetic Field Sensing

The low-frequency IDE sensors employed in capacitance measurement process their activities in a surface area of electro-quasi-statistic that is almost similar or close to Maxwell equations. The analog surface area of these sensors is referred to as magneto-quasi-static, also commonly known as meandering winding magnetometers [77]. A flow of electric current through the windings results in the induction of eddy currents. As the current passes from the primary winding voltage to the secondary winding voltage, the magnetic flux time rate of change is calculated. At low frequencies, the voltage becomes small and insignificant. Therefore, a magneto-resistive sensor is employed to act on behalf of the secondary winding voltage to overcome this challenge [78]. Low-frequency voltage is only required when dealing with testing materials that have flaws that are not visible [79]. Magnetometry sensors can also be employed in spectroscopy due to the ability to allow similarity in the current frequency with an external source.

Table 2 provides a comparison of various magnetic field sensors based on geometrical mean distance (GMR), range of magnetic field strength (H Range), and sensitivity revealing distinct characteristics that cater to different application needs. The GMR sensor demonstrates moderate sensitivity at 120 V/T and a GMR range between 10^12^ and 10^2^. On the other hand, the Hall sensor exhibits a sensitivity of 0.65 V/T, indicating an ability to detect weaker magnetic fields, coupled with a wide H Range spanning from 10^6^ to 10^2^ T. In contrast, the SQUID sensor stands out with exceptional sensitivity at 10^14^ V/T and a GMR range of 10^14^ to 10^6^, making it particularly well-suited for detecting extremely weak magnetic fields. The conclusions drawn underscore the importance of considering specific application requirements, as GMR and Hall’s sensors offer versatility across a broad range of magnetic fields, while SQUIDs excel in applications demanding sensitivity to the minutest magnetic field changes. The optimal choice among these sensors depends on the targeted sensitivity and the magnetic field strength range pertinent to the application at hand.

In summary, the choice of magnetic field sensor depends on the specific application’s requirements, including the desired sensitivity and the range of magnetic field strengths to be measured. GMR sensors and Hall sensors are suitable for a wide range of applications, while SQUIDs excel in detecting extremely weak magnetic fields.

### 6.2. Acoustic Sensors

The principle is focused on piezo acoustic interdigital transducers (IDTs), which it argues to be the main component of surface acoustic wave (SAW) devices often used in processing signals and measuring materials. The main trait of SAW devices is that their acoustic energy is stored on the surface of a solid [81]. Therefore, when a wave or current is moving across the surface of the solid, it can be trapped. Another important trait is their low velocity compared to that of electromagnetic waves. Therefore, their acoustic waves have low loss and fewer wavelengths, making them highly suitable for long delay lines. The building blocks of a SAW sensor are IDT which often, in almost every sensor, there are two IDT blocks [82,83]. On the surface of a solid is a single IDT photo deposited on it [84]. After applying a voltage, its distribution occurs between spatially periodic electrodes. The piezoelectric effect leads to the distribution of a strain in response to the wavelengths of the waves on the solid’s surface. Then the SAW sensor stretches in two directions, influenced by the IDT [85]. The second IDT detects the SAW sensors using the inverse coupling method of the piezoelectric. There are emerging technologies that explore the use of acoustic sensing for humidity measurements [86]. Recent research in the field of acoustic sensors has explored the integration of printed electronics and IDEs to enhance sensor performance. Utilizing flexible substrates and inkjet printing technology, researchers have developed lightweight and conformable sensors with precise patterning of conductive and piezoelectric materials such as silver nanoparticles and carbon nanotubes. The use of IDEs, known for their high sensitivity, allows for miniaturization and frequency selectivity, making these sensors suitable for applications where space constraints and specific frequency ranges are critical. The interdigitated pattern on electrodes enhances surface area and, when integrated with signal processing techniques, contributes to improved accuracy in detecting and distinguishing various sounds. This research has implications for diverse applications, including structural health monitoring, healthcare, and IoT devices, and underscores the continuous evolution of printed electronics and IDEs in advancing acoustic sensor technology.

## 7. Modeling of Transducers Based on Interdigitated Electrodes

The demand for flexible energy sources integrated with piezoelectric layers has surged with the advancement of reduced and wearable microelectronics [81,82]. The inherent polarization property induced by electric fields in flexible materials experiencing distortion finds application in various fields, particularly as energy transducers [83,84]. A prevalent approach to modeling energy transducers involves the utilization of IDE capacitors integrated into flexible polymeric substrates such as polyethylene, polypropylene, and ethylene vinyl acetate. These materials serve as passive electronic components and IDE structures offer notable advantages including ease of implementation, high sensitivity, and cost-effectiveness, beyond their primary function as capacitors [85]. IDE structures represent a widely adopted design in piezoelectric sensors.

The analytical evaluation of the interdigital electrode capacitance for a multilayered structure has been extensively studied in the literature. Igreja and Dias [87] developed an analytical model for the capacitance of interdigital electrodes in a multilayered structure, which was further extended in their subsequent work [88]. Additionally, Blume et al. [89] proposed a model for the capacitance of multi-layer conductor-facing interdigitated electrode structures.

### 7.1. Capacitive Devices

Using a sensor founded on an IDE structure is vital in the modeling of transducers. In this case, capacitive sensors are significant in the appropriate functioning of multifaceted electronic systems. A previous study established that capacitive sensors relate to the changes in capacitance passing through two or many conductors in a dielectric location [90,91]. Capacitive sensors are largely chosen due to their quick response [92] and their lower power consumption compared to resistive sensors. Modeling supercapacitors with interdigitated electrodes as shown in Figure 4 requires thought about the nontrivial geometry, this study takes into consideration conformal mapping, electrostatic equations, and finite element methods to gain insights into the reliance of interdigitated cell’s electrical behavior of the electrodes on its geometric features [93,94].

### 7.2. Resistive Devices

The demand for reduced and wearable microelectronics, along with the necessity for flexible energy sources integrated with piezoelectric layers, has driven the exploration of resistive devices. As part of this, the modeling of transducers based on IDEs becomes crucial. Flexible polymeric substances, such as polyethylene, polypropylene, and ethylene vinyl acetate, commonly used in passive electronic devices, serve as ideal materials for IDE capacitors. The unique polarization property induced by electric fields through flexible distortion is harnessed for energy transduction purposes. IDE structures, characterized by their ease of use, high sensitivity, and cost-effectiveness, extend beyond their role as capacitors. Capacitive sensors, integral to multifaceted electronic systems, play a significant role in the appropriate functioning of these systems. Capacitive sensors are preferred over resistive sensors due to their quick response and lower power consumption. Modeling supercapacitors with interdigitated electrodes involves addressing the nontrivial geometry, and this study employs conformal mapping, electrostatic equations, and finite element methods to gain insights into the dependence of the electrical behavior of interdigitated cells on their geometric features. This comprehensive approach enhances our understanding of resistive devices based on IDEs, particularly in the context of capacitive sensors, contributing to the advancement of flexible and wearable microelectronics.

### 7.3. Conformal Mappings

They are employed to define the capacitance and impedance of interdigitated electrodes in multi-layered structures. For instance, based on conformal plotting, the overall capacitance of a multilayered IDE assembly is acquired through the addition of each layer’s contribution. That is, in the case whereby the electrodes are arranged on a substratum of limited thickness and a with no capping layer on top [95,96], the overall capacitance is found by:(1)$$C=2C_{air}+\left(\varepsilon_{sub}-1\right)C_{sub}$$
where 2C_air_ is the “influence from the substantially impenetrable layers of midair above and under the IDE structure with a dielectric constant 1” [97,98]. Further, C is the substrate with the dielectric constant.

In addition, the conforming mapping equation is used to obtain the effect of the ratio of electrodes on the capacitance spacing for interdigitated [99]. Following these assumptions, the total capacitance of a device can be given by:(2)C=NLC
where N is the number of layers midair above and under the IDE structure and L is the length.

Considering the above equations, compared to numerical analysis, one key advantage of conformal mapping is the degree of precision to which cell capacitance values can be calculated [100,101]. Conformal mapping is highly applicable in developing effective solutions for the Laplace equation on sophisticated devices and subjects such as aerodynamics and elasticity [101].

The close link connecting complex analysis and Laplace equations stimulates one to view the need for complex functions in defining variables. In this regard, a complex analytic function can be defined as mapping by using the below equation [102].
ζ = g(z) or ξ + iη = p(x, y) + iq(x, y)(3)
where ζ: This represents a complex number. In the context of your equation, ζ is the complex variable being defined in terms of real variables x and y, along with complex constants α and β. It is typically used to represent a point in the complex plane, where ζ = ξ + iη, with ξ and η representing the real and imaginary parts of the complex number, respectively.

g(z): This represents a function of a complex variable z. In other words, it is a mathematical function where the input (z) and output (g(z)) are complex numbers. For example, g(z) could be any function that takes a complex number z as input and returns another complex number as output.ξ: This represents the real part of a complex number. In the equation ζ = g(z) or ξ + iη = p(x, y) + iq(x, y), ξ is the real part of ζ.η: This represents the imaginary part of a complex number. In the equation ζ = g(z) or ξ + iη = p(x, y) + iq(x, y), η is the imaginary part of ζ.p(x, y): This represents the real part of a complex-valued function. In the context of complex analysis, complex-valued functions are often expressed as a sum of a real part and an imaginary part. Here, p(x, y) represents the real part of such a function, where x and y are real numbers.q(x, y): Similar to p(x, y), this represents the imaginary part of a complex-valued function. In complex analysis, complex-valued functions are often expressed as a sum of a real part and an imaginary part. Here, q(x, y) represents the imaginary part of such a function, where x and y are real numbers.z: This represents another complex number. It is typically used as the input variable for the function g(z). z = x + iy stands for a prescribed domain

Ω ⊂ C to a point ζ = ξ + iη to stand for the image domain D = g(Ω) ⊂ C

In many instances, the image domain D stands for the unit disk, but as demonstrated in Figure 5 below:

Any simply linked domain in the complex plane, excluding the plane itself, is conformally equal to the unit disk, according to the well-known Riemann mapping theorem. It can be time-consuming to locate an explicit conformal map for a particular domain, though. Every properly connected simply connected subdomain of the complex plane is conformally equal to the unit disk by the Riemann mapping theorem. For the majority of the 20th century, the study of conformal mappings on the unit disk gave rise to the traditional field of mathematics known as univalent functions. Plotting multiple conformal maps defined on the unit disk is done in this demonstration. It is standard practice in this field to normalize each map so that each plot shows radial lines emanating from the origin and concentric circles encircling it.

Some of the basic analytic mappings applicable in providing a solution to solve Laplace equation value problems are:ζ = z + β = (x + a) + i(y + b)(4)

With β = a + ib being a complex fixed number. The result is a translation of the entire complex plane in line with the direction provided by the (a, b) T vector [103].

To obtain efficient packaging on a thin substrate, there is a need for extensive exercise that will ensure that no excess stress is exerted during the assembly process, which often occurs due to the change in temperature and pressure. Finite element analysis (FEA) is applied in the modeling process to evaluate static analysis and minimize the bending and temperature effect. Four conditions are considered when applying the FEA process: shear, tensile, three-point bending, and temperature load [104]. The process allows optimizing the substrate layer, resulting in minimized stress.

### 7.4. Electrostatic Equations

On a parallel plate, the capacitance is in inverse proportionality to the surface area or distance separating the two plates. The general equation is provided as
(5)$$C=\frac{\varepsilon \left(A\right)}{d}$$
where ε is the absolute permittivity of the materials being measured.

The dielectric constant can also be represented as 1/(4π∙9 × 10^9^). Simplifying, an insulating solid is used to separate the parallel plates [86,105]. Therefore, when calculating capacitance, the permittivity of air is considered to have the same value as the permittivity of a vacuum.

The following equation is used to find the capacitance for an electrical cell that contains two conducting terminals partitioned by some dielectric or conductive medium of permittivity ε or conductivity σ [106]:(6)$$C=\frac{|\iint_{s}\epsilon \vec{E}\cdot \vec{dS}|}{\Delta V}$$

In this equation, ∆V is the difference in voltage between two terminals, Ē is the electric field, and S is the surface of one of the terminals.

#### Finite Element Methods

Furthermore, the capacitance of IDE sensors can be calculated using the finite element methods (FEM). The FEM-oriented arithmetic method solves Gauss’s law for an electric field by employing the specified amount of electric power as the independent parameter [107,108]. Thus, the capacitance of the design of IDE sensors can be calculated through two methods, which are the following.

Calculation following the terminal charge:(7)$$C=\frac{NQ}{2U_{0}}$$

Calculating on the foundation of the surface integral of the electric energy kept, we use
(8)$$C=N\frac{2}{U_{0}^{2}}\int_{\Omega }W_{e}d\Omega 2$$

In the above equations, C denotes the capacitance of the system, N is the number of electrode fingers, Q is the total charge, U_0_ is the applied possible difference between the ground and terminal electrode, and Ω is the enclosed 2D model surface [109].

Assembling capacitive affinity biosensors is possible through recognition elements, such as antibodies, which are finely layered and structured on the surface of electrodes. An IDE finger pattern is then employed to offer a higher surface area. The equation is [110].
(9)$$C=\varepsilon_{r}\varepsilon_{0}\frac{A}{D}$$

C represents the capacitance of the capacitor, where $\varepsilon_{r}$ is the relative permittivity (or dielectric constant) of the material between the plates. The parameter ε_0_ is a constant referred to as “permittivity of the free space”, with a constant value of 8.854 × 10^−12^ F/m. *A* represents the electrode plate surface area, while d represents the distance. According to the equation, a difference in the capacitance directly impacts the distance between the two plates and can adjust the overlapping area and dielectric permittivity. This equation is founded on the principle that the IDE capacitor is on the capacitive sensors. The formula has been proven correct by A Qureshi et al. and Trout et al. in their experiments, which demonstrated that the distance between the plates changed with a change in capacitance [111,112].

When a force is applied, the capacitance change in a material is attributed to the material’s flexibility. Two parallel plates can be employed to approximate a sensor. When the distance between the parallel plates is less than the distance between the electrodes (6.4 mm × 36 mm versus 0.2 mm), the approximation of the capacitance sensor can be calculated [113]. According to generic thermoplastic polyurethanes, 0.5 is the closest ratio of polyurethane width and length. Therefore, increasing the compression of the dielectric area of the electrode for volume to be (V) will not influence any change in the polyurethane resulting in,
(10)$$C=\varepsilon_{r}\varepsilon_{0}\frac{A}{D}=\frac{\varepsilon_{r}\varepsilon_{0}V}{d2}$$

C represents the capacitance of the sensor, where $\varepsilon_{r}$ is the relative dielectric constant of the dielectric material, $\varepsilon_{0}$ is the vacuum permittivity, A is the surface area of the plates, d represents the distance between the plates, and V is the voltage applied across the plates. Equation (10) represents the formula for capacitance, which is derived from the formula for the electric field between parallel plates with a dielectric material inserted between them.

Less significant changes on an initial area A_0_ with an initial thickness of d_0_ will lead to a capacitance change from the partial derivative of C with respect to d:(11)$$C=\frac{\frac{\partial C}{\partial D}}{d}\to d_{0};∆d=-2\frac{\varepsilon_{r}\varepsilon_{0}A_{0}}{d_{0}}∆d$$

It is important to note that in an incompressible dielectric, the change in capacitance could be more than double the compressible dielectric due to the change in the area between the plates. In a case where the material used is in the linear range, the equation to be used is:(12)$$∆d=-\frac{Fd_{0}}{A_{0}E_{1}}$$

In this equation, F is the change in capacitance and E_1_ is the change in electric energy.

With their many features and complementary abilities, electrical and electrochemical sensors are highly valuable instruments for the identification and examination of analytes in a wide range of applications. These characteristics—which include detection limit, reaction time, stability, sensitivity, and selectivity—are frequently backed by data taken from the specifications of commercial sensors. Commercial sensors offer useful information about operating parameters, calibration techniques, and performance metrics. This information sheds light on the strengths and weaknesses of the sensors. Even so, there is still a lack of knowledge on the characteristics and complementary abilities of these sensors, especially in theoretical and practical settings.

The ability to multiply electrodes without appreciably raising costs is one noteworthy feature of electrical sensors, especially IDE sensors. IDE sensors offer improved sensitivity and spatial resolution thanks to their comb-like arrangement of numerous electrodes. While adding additional electrodes could seem like a good idea, it is important to take into account the reasons why some sensors choose to use more than two electrodes, even though there is not much of a financial impact [57].

There are other considerations besides price when choosing to employ several electrodes in sensors. Instead, the need for improved reliability and performance drives it. By adding more electrodes, sensors can decrease the effects of electrode degradation, decrease signal drift, and increase overall resilience. Additionally, by permitting redundancy in signal collecting, the use of many electrodes minimizes the likelihood of false readings and boosts sensor accuracy. Additionally, the use of more than two electrodes allows differential readings, which can provide crucial information on the gradients and distribution of analyte concentration in the sample. In conclusion, the reasoning behind using numerous electrodes in some sensors extends beyond economic concerns, even though the specifications of commercial sensors provide invaluable help for comprehending sensor features. These sensors offer improved performance, accuracy, and dependability by utilizing electrode multiplication, which makes them invaluable instruments for a variety of electrical and electrochemical sensing applications [56].

### 7.5. Optimization of IDE

Electrode geometry is used in the analysis of capacitance. The capacitance and impedance can be changed and adjusted per the active area of an IDE [114,115]. For instance, in medicine, the capacitance biosensors have been proven to demonstrate that in cancerous breast tissue, the percentage of the differential impedance is an approximated 6.9% more when employing a 10 μm compared to when using a 30 μm IDE space area [116]. This shows that when the IDE area gap is increased, it increases its impedance [117]. Therefore, capacitance and impedance are inversely proportional; a decrease in impedance increases capacitance on the IDE surface area. Therefore, the effective electrode area is increased by minimizing the electrode space, resulting in increased cell sensitivity [118,119]. In the optimization of IDEs, several electric values are considered to achieve the desired performance characteristics. These values are typically adjusted and optimized to enhance the sensing capabilities of the IDE-based sensor. Table 3 shows electrode material and conductivity values used in the optimization of IDEs.

### 7.6. Advantages and Disadvantages of the Modelling Methods

The advantage of using FEM is that it accommodates a variety of material properties in modeling. The method also allows the implementation of higher-order elements, and it is simple, compact, and result-oriented [121]. Disadvantages of FEM include demands for large input data, prolonged execution time, and variations in output results [122].

The advantage of the electrostatic equation in the calculations of capacitance is that the equation involves the basic quantities that affect capacitance directly; it is, therefore, compact [123]. In the process of calculating capacitance, the user can also find the value of other relevant quantities as a bonus to the calculation. The disadvantage, therefore, is that the calculation process is quite complex and even tiresome. Using this method, the user is forced to find other quantities that they never intended to calculate in the first place [124]. The analysis is theoretically accurate.

While the current fabricated technologies show great potential for meeting the needs of printed and flexible electrodes, there are many drawbacks, including the high costs, low solution, and low-conductivity manufacturing activities. For instance, in conventional fabrication processes, there is a need for high-temperature treatment for evaporation and the metal to be deposited. The printed technology needs a sinister activity for high conductivity. However, for these methods to occur efficiently and successfully, a high cost must be incurred, particularly in thermal treatment [125]. Therefore, industrial and mass production need to consider the above limitations.

Some of the challenges of printed and flexible electronics include the need for low-cost and high-performing materials with effective electrolyte-leading [125]. Currently, the operating high-efficiency materials can only run for a lifetime of a few 1000 h. Therefore, only low blue-emitting fluorescent materials are used in organic light-emitter diodes (OLED) production, presenting a major challenge in electronics.

## 8. Characteristics of Resistive Devices

### 8.1. Resistors and Transistors

In resistors and transistors, the *W*/*L* ratio (width-to-length ratio) represents the amount of current flow. This means that increasing the ratio *W*/*L* increases the available current. Considering that resistance is inversely proportional to available current, reducing *W*/*L* increases resistance. Further, when the *W*/*L* of a transistor is increased, the transistor is provided with more current drive capacity. When exposed to sensing technologies, the properties of capacitive and resistive technologies are demonstrated in Table 4 below.

Interdigitated metal electrodes are extensively used for different sensor applications, including humidity sensors, biosensors, and gas sensors, among others [126]. IDE sensors that occur in solid-state electrochemical devices are used in temperature and conductivity detection. They are also used to detect the dielectrical properties of a sample. Common properties of these sensors include real-time response rate, adaptability to miniaturized systems, low cost, and, therefore, being suitable in mass production, high reliability, and low impedance output signal. Therefore, they are considered applicable in special devices that are adapted to any part of process equipment, noble metal electrodes, and automation of packages, among other technologies.

### 8.2. Chemical Properties

Previous research proposed a new fabrication strategy intended to enhance the value of capacitance per surface area of IDE devices inkjet printed on foil in a dependable way [126]. The strategy encompasses the deposition of a thin dielectric layer of parylene-C onto the first printed electrodes before the printing of the second comb [127].

The results showed that the introduction of a dielectric interlayer to the fabrication strategy is valid for the use of complex inkjet printers, especially where short circuits cause a device to be non-functional [128]. In addition, the fabrication strategy allows for the enhancement of capacitance per surface area.

IDE devices’ chemical characteristics can differ significantly depending on the method used for surface functionalization and the intended use. The following are some typical chemical characteristics linked to IDE devices:Surface Functionalization: Chemical modifications can be applied to the surface of IDE devices to add particular molecules or functional groups. In applications like sensing and biosensing, this functionalization is crucial in deciding how the device interacts with target analytes or biological entities [17].Electrode Material Composition: The chemical characteristics of IDE devices can be greatly impacted by the selection of electrode material. The degrees of chemical stability, reactivity, and environmental compatibility differ amongst materials. For instance, in biomedical applications, noble metals like gold or platinum are frequently selected due to their inertness and biocompatibility [129].Chemical Reactivity: When IDE devices are in contact with analytes, electrolytes, or other elements in the surrounding environment, chemical reactions may occur. In uses like chemical sensing or electrochemical energy conversion, the degree of chemical reactivity can affect the device’s performance, sensitivity, and selectivity [130].Surface Wettability: The adsorption and transport of molecules or ions to the electrode interface are influenced by the wettability of the electrode surface, which is determined by its chemical composition and surface energy. In applications such as electrochemical sensing or microfluidic systems, the device’s performance can be improved by controlling surface wettability through chemical modification [15].Biocompatibility: Biocompatibility is essential in biomedical applications to ensure that there are as few negative consequences as possible when integrating IDE devices with biological systems. Enhancing biocompatibility and lowering the risk of an immunological reaction or cytotoxicity can be achieved through surface changes or coatings.Corrosion Resistance: In electrochemical applications, in particular, IDE devices may be subjected to corrosive conditions. For devices to remain functional and intact over time, surface coatings and electrode materials must be resistant to chemicals.

Overall, understanding and controlling the chemical properties of IDE devices are essential for tailoring their performance to specific applications, whether in sensing, energy conversion, biomedicine, or other fields. By leveraging surface functionalization and material selection, researchers can optimize IDE devices to meet the demands of diverse and challenging environments. Notice that the chemical properties of IDE devices are often tailored and optimized for specific applications. The choice of electrode materials, surface modifications, and functionalization methods can significantly impact the device’s chemical properties and its ability to sense and analyze specific chemical compounds or properties in the environment [56]. The potential of the fabrication of interdigitated capacitive chemical sensors was demonstrated through the functionalizing of the IDE structure with a humidity sensing layer and the subsequent sensor against relative humidity.

#### 8.2.1. Humidity Detection

Electronics industries use humidity sensors to control relative humidity during manufacturing. Relative humidity (RH) sensors are grounded on ceramic constituents that include aluminum oxide and semiconductor constituents including polymers and SiO_2_ [128].

A previous study carried out a comparison between two evolving fabrication techniques involving inkjet printing and nanographene creation for the growth of conductive designs of elastic electronics [131]. The inkjet printer manufactures silver-based electrodes, while the laser scribing technique produces laser-induced graphene (LIG) and laser-induced graphene oxide (laser-rGO) designs from graphene oxide (GO). These methods are employed to form planar IDE capacitors. Moreover, the comparison concentrates on the use of IDE capacitors as RH sensors. In terms of sensitivities to RH, inkjet-printing technology demonstrates a reduced value of capacitance and better-quality performance as a capacitive structure [132]. Thus, factory-made capacitors depict a highly viable performance as capacitor constructions.

When IDE structures are used as humidity sensors, the sensor starts with a capacitive performance, and while the temperature increases, it becomes resistive. Furthermore, the change in the parallel resistance with respect to the frequency is due to the leakage of current and the dielectric losses as temperature increases. This behavior can be attributed to the combined effects of current leakage and dielectric losses associated with changes in temperature [133,134].

At lower temperatures, the IDE structure behaves primarily as a capacitive sensor for humidity measurement. This is because the humidity-sensitive material or coating applied to the electrodes exhibits a dielectric response to changes in humidity. When humidity levels change, the dielectric properties of the material, such as its relative permittivity, alter, leading to changes in the capacitance of the IDE structure. The variation in capacitance is then correlated to the humidity level.

As the temperature increases, the IDE structure undergoes a transition from a capacitive response to a resistive response. This change is primarily attributed to two factors: current leakage and dielectric losses. At higher temperatures, the leakage current between the interdigitated electrodes increases. This leakage current can arise from various sources, such as increased ion mobility within the humidity-sensitive material or changes in its electrical conductivity. The increased current leakage leads to a higher resistive component in the IDE structure’s response. As temperature rises, the humidity-sensitive material may exhibit increased dielectric losses, also known as loss tangent or dissipation factor. Dielectric losses refer to the energy dissipated in the material as it undergoes polarization changes under an applied electric field. These losses contribute to the resistive component of the IDE sensor’s response. The change in parallel resistance with respect to temperature frequency arises due to the combined effects of current leakage and dielectric losses. As the frequency of the applied electrical signal changes, the response of the IDE structure may vary.

At higher frequencies, the capacitive reactance dominates the sensor’s response, resulting in a lower parallel resistance. In contrast, at lower frequencies, the resistive component, including the effects of current leakage and dielectric losses, becomes more prominent, leading to an increase in the parallel resistance. A comprehensive explanation of the behavior of IDE humidity sensors has been presented in relation to current leakage, dielectric losses, and temperature. The capacitive performance of the IDE structure at lower temperatures, the transition to a resistive response with temperature increase, and the change in parallel resistance with frequency are all key aspects to consider in understanding the behavior of such sensors.

It is important to note that the exact behavior and characteristics of IDE humidity sensors can vary depending on the specific design, materials used, and environmental conditions [86]. The mentioned capacitive-to-resistive transition with temperature is a common observation, but the extent and details of this transition can differ for different sensor configurations and humidity-sensitive materials. It is always essential to consider these factors when analyzing the behavior of humidity sensors and their response to temperature changes.

#### 8.2.2. Gas Detection

High sensitivity and selectivity are desired for the gas detector using the inkjet printing technique.

Considering a previous study whereby IDEs for gas sensors were fabricated through inkjet printing technology [133], results show that: The resistance of IDEs rapidly increases with the rising number of printed layers (p/p^o^) from 1 to 4 layers. Due to the high sensitivity and high selectivity of the gas sensor, it was able to detect the presence of ammonia gas. Normally, resistance should reduce linearly in relation to the number of printed layers. Therefore, these results demonstrated nonlinearity. It involves the identification and quantification of specific gases in a given environment. The limit of detection (LOD) is a critical parameter indicating the lowest concentration of a gas that a detection system can reliably identify. The LOD is often influenced by factors such as sensor technology, measurement techniques, and environmental conditions.

In previous studies related to gas detection, the LOD has been a focal point for researchers. The LOD can vary widely based on the type of gas sensor employed, ranging from traditional electrochemical sensors to more advanced technologies like semiconductor sensors or optical sensors. Each sensor type has its own advantages and limitations in terms of sensitivity and selectivity.

The conditions used in these studies depend on the research objectives and the nature of the gases being investigated. Researchers often expose sensors to controlled concentrations of gases in laboratory settings to assess their response characteristics. The conditions may involve varying factors such as temperature, humidity, and the presence of interfering gases to simulate real-world scenarios.

Some studies focus on the detection of single gases, while others explore the response of sensors to mixtures of gases. Understanding the sensor’s selectivity, or its ability to differentiate between different gases, is crucial for practical applications. Researchers aim to enhance selectivity to ensure accurate detection in complex environments where multiple gases may be present simultaneously.

To gain detailed insights into the LOD, experimental setups, and selectivity of gas sensors, it is advisable to refer to specific research papers in the field. The methods and results sections of these papers typically provide comprehensive information on the experimental conditions, sensor performance, and the significance of the findings.

A gas detector is a device or sensor used to monitor and detect the presence of gases in the inkjet printing environment. Inkjet printing techniques involve the deposition of ink onto a substrate, and certain gases emitted during the printing process can have implications for safety, print quality, or the performance of the printing system. Gas detectors are employed to ensure a safe working environment, maintain print quality, and prevent any adverse effects caused by the presence of harmful or undesirable gases.

Here are some key points regarding the gas detector’s role and functionality in inkjet printing:(i) Gas Monitoring: Gas detectors continuously monitor the air or gas composition in the printing environment. They detect and measure the concentration of specific gases of interest that may be emitted during the printing process, such as volatile organic compounds (VOCs), ozone, ammonia, or solvents.(ii) Hazard Detection: Gas detectors are crucial for identifying and alerting operators to the presence of hazardous gases in real time. If certain gases exceed safe concentration levels, the gas detector can trigger alarms or shut down the printing system to prevent harm to the operators or equipment.(iii) Air Quality Control: Maintaining good air quality is essential for achieving optimal print results. Some gases emitted during inkjet printing, such as VOCs, can affect the print quality, ink adhesion, or drying time. Gas detectors help to ensure that the air quality is within acceptable limits for achieving the desired print outcomes.(iv) Calibration and Specific Gas Detection: Gas detectors in inkjet printing systems are often calibrated to detect specific gases relevant to the printing process. The calibration ensures accurate and reliable gas measurements specific to the target gases of concern.(v) Integration with Printing System: Gas detectors can be integrated into the inkjet printing system, either as standalone devices or as part of a centralized control system. They can communicate with the printing system’s software or control unit to trigger appropriate actions, such as adjusting printing parameters, activating safety measures, or generating alerts.(vi) Maintenance and Regular Checks: Gas detectors require periodic maintenance and calibration to ensure their continued accuracy and reliability. Regular checks and calibrations help to guarantee that the gas detector is functioning correctly and providing accurate gas concentration readings.

By incorporating gas detectors into inkjet printing systems, operators can proactively monitor and manage the presence of gases emitted during the printing process. This ensures a safe working environment, helps maintain print quality, and mitigates any potential risks associated with harmful gases.

In terms of sensor performance for ammonia (NH_3_) detection, the resistance of polymer chips increases with the availability of NH_3_ vapor [134]. Gas sensors deliver a resistance of high value, unlike in the case of capacitance. Considering that the poly(3,4-ethylenedioxythiophene) doped with polystyrene sulfonated acid, PEDOT/PSS (Poly(3,4-ethylenedioxythiophene)-poly(styrenesulfonate) was used as a sensing film in a previous study, (PEDOT/PSS) gas sensor shows high sensitivity and selectivity to ammonia gas. The PEDOT/PSS constituent demonstrates a strong response to NH3 and its response functions linearly with gas concentrations between 100–1000 ppm. The results depicted in Figure 6 show that IDEs fabricated by the inkjet printing method could be used as a sensor platform for gas detection.

#### 8.2.3. Biomarkers Detection

In the existing literature that focused on the structure, formation, and features or behaviors of a biosensor for medical applications, the results revealed that the capacitance value increases when a buffer solution is used as a dielectric fluid, and, moreover, in a case where the conductivity is increased by decreasing resistivity. In summary, the use of a buffer solution as a dielectric fluid can enhance capacitance in a biosensor, while the use of metals and the decrease in resistivity can increase conductivity, leading to improved electron flow and sensitivity in detecting changes relevant to medical applications. These features are crucial for the performance and reliability of biosensors in various diagnostic and monitoring applications in the medical field. Most importantly, bio-molecule functions as a dielectric and decreases the gap size between electrodes [133]. Aluminum is considered appropriate for biomolecule detection. nature-inspired chemical sensors for enabling fast, relatively inexpensive, and minimally (or non-) invasive diagnostics and follow-up of health conditions. Biosensors are analysis tools that mix the detection of a biomolecule and are applied settings for determining biomarkers within the biofluids. Many physical sensors based on optical properties sense physical biomarkers that sense mechanisms for selection and detection.

In a biosensor designed for medical applications, the choice of a dielectric fluid, such as a buffer solution, plays a crucial role in influencing capacitance and conductivity. The dielectric effect, governed by the dielectric material between the capacitor plates, becomes evident when a buffer solution is employed in the biosensor setup. This choice enhances capacitance due to the increased dielectric constant (k) of the buffer solution, impacting the capacitor’s ability to store charge. The capacitance (C) is determined by a formula directly proportional to (k) and the surface area (A), while inversely proportional to the distance between the plates (d). Simultaneously, the conductivity of the biosensor, vital for its electrical response, is heightened using metals with high electrical conductivity. This increase in conductivity, achieved by decreasing resistivity, facilitates enhanced electron flow through the biosensor material. The amplified electron flow, a consequence of lowered resistivity and the incorporation of metals, contributes to the biosensor’s heightened sensitivity in detecting changes induced by biological interactions, a crucial aspect in medical applications.

### 8.3. Physical Properties of Resistive Devices

This section involves the description of the previous works in printed sensors and other materials, as well as the plotting of resistance against temperature. It will also explore the strain and the impact of strain on the generated signal. The curves represent a drop in the sensor resistance with the increase in temperature. Samples with low humidity demonstrate high reaction compared to the formation procedure.

#### 8.3.1. Temperature

Previous works describe the generation of inkjet printer sensing devices for temperature on a bendable polyethylene substrate and found that different sensors show a linear response to temperature and that as sensor resistance decreases, temperature increases [97,104,119]. In this literature, the authors clearly illustrate materials used in the research were polymer-based poly(3,4-ethylenedioxythiophene) polystyrene sulfonate (PEDOT: PSS) ink, and silver nanoparticle (AgNP) ink. The temperature sensors are primarily capacitive above a frequency of 1 MHz.

Figure 7 demonstrates the sensor response for various fabrication flows and electrode spacing of 200 μm. Moreover, the plots comprise data from the different processes that are referred to as dry, wet, and sintered. The curves depict a decrease in sensor resistance as temperature increases. Dry samples demonstrate a higher response compared to the other formation processes. In addition, the study provided a summary of the performance parameters of the PEDOT: PSS-based sensors used, along with the substrate material, deposition technique, and composite sensing materials.

#### 8.3.2. Strain

Strain sensors are commonly used mechanical sensors that are used to measure mechanical deformation. A previous study that investigated the impact of strain on the produced signal of very stretchy interdigitated capacitive (IDC) sensors established that the comparative variation in capacitance of the IDC sensor depends on the strain and independent of transducer dimensions that include length, thickness, and spacing of electrodes [101]. Additionally, strain sensors function at higher rates of strain of up to ~500 mm/min in actual applications, as shown in Figure 8. The impact of strain rate on the non-linear elastic behavior is smaller at a lower filler loading. Thus, the stress and strain relationship as well as the non-linear property of the composite reduces with the rise in matrix filling load.

In the study, the highly stretchy IDC sensor was made using barium titanate (BTO)-silicone elastomer composite [116]. Another material that was used was the carbon black/Eco flex conductive ink that used a custom 3D printer [110]. The resulting voltage from the sensors does not change, but there is a relatively high variation seen in some samples. The variation may be caused by non-uniform sampling of the piezoelectric components across the IDE section. Further, because of the bent result of the IDC sensor, the gauge factor (GF) of the IDC sensor reduces with the strain.

## 9. IDE for Printed Field Effect Transistors

The significance of field-effect transistors (FETs) and stretched gate field effect transistors (EGFETs) in printing technology and integrated circuits (ICs) is underscored by their compact size and low power consumption. These sensing techniques, involving regulated modifications in the electric field within the local channel, operate through the binding of charged molecules onto nanowire surfaces. FETs, integral in the semiconductor electronics industry, are key in the production of various electronic devices such as gate insulators and voltage variable resistors (VVRs) in operational amplifiers (OP-Amps). While modern FETs often rely on inorganic materials using conventional manufacturing methods, an alternative approach involves ink prints, enabling mass production with cost-effective resources. In parallel development, advancements in materials science and the demand for innovative functionalities in electronic devices have fueled progress in organic electronics. This evolution is supported by efforts to achieve efficient manufacturing processes, including low-cost, high-volume printing techniques. Organic electronic semiconductors, pivotal for flexible devices, offer advantages like the processability of constituent elements for diverse printing needs, low-temperature fabrication, mechanical flexibility, and the ability to coat large areas. The integration of polyelectrolytes as gate insulators enhance performance by enabling low-voltage operation within 1 V, preventing unintentional electrochemical doping of the organic semiconductor bulk, and ensuring tolerance to thick gate insulating layers and proper electrode alignment along the channels. This convergence of FETs, EGFETs, and organic electronics reflects a dynamic landscape in electronic device manufacturing, with implications for enhanced functionality and efficient production processes.

## 10. IDE-Based Energy Devices

The study of piezoelectric properties of the IDE-based sensors leads to a change in the size of the thickness of the layers. Additionally, the output voltage of the sensors remains constant. Relatively great ¡variation might be caused in samples of printed materials by non-uniform sampling of the piezoelectric components across the IDE section. High piezoelectric sensitivity is observed on the sample set with the lowest thickness. The field of IDE-based sensors is highly dependent on the thickness of P (VDF-TrFE) layers.

The mechanical stability of IDEs is important in the context of energy devices, especially in piezoelectric and triboelectric harvesters, because IDEs are used in flexible and wearable electronics. IDEs are essential to this energy harvesting device’s ability to gather charge carriers effectively. For example, in piezoelectric energy harvesting, IDEs maximize the extraction of generated electric charge by offering an organized electrode layout, which improves charge-collecting efficiency. Comparably, IDEs help create a clear channel for charge carriers to travel in the direction of their respective electrodes in triboelectric energy harvesting, which reduces charge loss via recombination. IDEs also help to minimize resistance inside the device and optimize electrode design to minimize energy losses and improve electrical performance overall [135].

The mechanical stability of IDEs is greatly influenced by the selection of electrode materials and patterns, especially when flexible electronics are involved. Research on printed polymeric flexible energy harvesting elements is essential because it clarifies how electrode designs and materials affect device durability, particularly regarding IDE design. By taking care of these issues, scientists may create durable IDE-based energy harvesting devices that are appropriate for flexible and wearable electronics applications. This will ultimately advance energy harvesting and help create sustainable energy solutions [89].

### 10.1. Supercapacitors and Batteries

Supercapacitors and printed batteries are both made up of electrolytes and electrodes. Batteries are praised for their high energy density but are also criticized for their limited life cycles and disposal issues [136]. Supercapacitors have longer life cycles but are limited by their DC (direct current) voltage incapacity to last for long and low energy density.

IDE has been extensively incorporated in the fabrication of batteries and super-capacitates due to IDE’s high-end features, including portability, durability, and flexibility [21]. In these energy devices, several cells are connected in series, with each cell in the series having its voltage potential, which derives the total terminal voltage. Higher capacity is attained in parallel connected cells summing up the total ampere-hour (Ah).

In some energy devices, the cells may be organized in series and parallel forms. An example is the battery of a laptop, which often includes four 3.6 V Li-ion cells in a series connection to add up to the nominal voltage, a parallel connection of cells with a capacity spanning from 2400 mAh to 4800 mAh [113]. Weaker cells in an energy device result in an imbalance; therefore, it is important that the battery or supercapacitor has equal voltage and capacity (Ah).

Being an alternative to batteries, supercapacitors mainly store energy electrostatically. Because of superior power density, cycle lives of batteries, and stability, supercapacitors (SCs) are significant energy storage devices for future electronic systems. Printed devices represent a paradigm shift of SCs as they offer a lot of simple, inexpensive, multipurpose, timesaving, and environmentally friendly manufacturing technologies for SCs that have new and desired structures, thus unleashing the potential of SCs for future electronics [136]. The emerging printed SCs that include inkjet printing, screen printing, three-dimensional printing, and transfer printing, among others, inspire the manufacturing of electronic devices at decreased fabrication and prototyping costs and permit the realization of devices.

Printed batteries are batteries for which at least one of the elements is solution-produced and deposited via printer. The architecture and design of printed batteries determine their mechanical features. In the modern era, printed sequences are widely established through 3D printing machinery. In fact, in the recent past, several research groups have attempted to fabricate a 3D lithium-ion series of 3D printing because of their decreased cost and convenience.

### 10.2. Solar Cell Tracking and Optimizing Devices: Enhancing Solar Energy Harvesting

Solar cell tracking and optimizing devices are crucial components of modern solar energy systems, designed to maximize the efficiency and output of solar photovoltaic (PV) cells. These devices play a pivotal role in harnessing sunlight effectively and ensuring that solar panels capture the maximum amount of energy throughout the day. In this discussion, we will explore the significance, types, and benefits of solar cell tracking and optimizing devices.

IDEs are crucial components within solar cells, particularly in the domain of solar cell tracking and optimizing devices, contributing significantly to their efficiency and performance. The specific roles of IDEs in this context encompass various aspects. Firstly, IDEs are designed to enhance charge collection efficiency by providing a structured and closely spaced arrangement of electrodes, facilitating the efficient collection of generated electrons and holes, and maximizing overall charge extraction from photogenerated carriers. Additionally, IDEs play a pivotal role in reducing series resistance within solar cells, optimizing electrode design to minimize the distance electrons need to travel before collection, thereby mitigating energy losses and enhancing the overall electrical performance of the solar cell. Moreover, IDEs contribute to the efficient separation of photogenerated carriers in the semiconductor material, offering a well-defined pathway for charge carriers to move towards their respective electrodes, preventing recombination, and improving the likelihood of collection. IDEs are also instrumental in supporting advanced solar cell designs, particularly in tandem or multi-junction configurations, where precise control over charge separation and collection is paramount for achieving desired electrical performance. Finally, IDEs contribute to improving the fill factor, a key parameter influencing solar cell efficiency, by minimizing resistive losses and maximizing charge collection efficiency. In summary, interdigitated electrodes in solar cells, particularly in the realm of solar cell tracking and optimizing devices, are tailored components crucial for enhancing charge collection, reducing resistance, facilitating carrier separation, and supporting advanced solar cell designs, ultimately contributing to the overall efficiency and output of solar energy systems.

## 11. Conclusions and Future Perspectives

This paper has examined the function of IDEs in printed and flexible electronics. The findings show that IDEs are widely applied in the production of sensors because of their fewer fabrication steps and their ability to generate high output voltage. IDE-based sensors and structures have involved a lot of consideration in the field of printed and flexible electronics due to their ability to generate a high output voltage. Their wide use in sensor applications can also be attributed to their wide range of advantages, including material imagining capability, the convenience of the application of sensitive coatings, and the potential for spectroscopy capacities via variation of frequency of electrical excitement. Future developments in additive manufacturing and nanomaterials will make printed IDEs indispensable in a wide range of innovative applications. Expertly crafted by methods like inkjet or 3D printing, IDEs will consist of cutting-edge substances like carbon nanotubes, nanowires, or conductive polymers, providing remarkable conductivity and longevity. With their great sensitivity and low cost, these printed IDEs will be very useful in areas such as energy storage, environmental monitoring, and biological sensing. They will make it possible to create high-performance energy storage devices for portable gadgets, wearable biosensors for continuous health monitoring, and smart environmental sensors for pollution detection.

The role of IDEs in printed and flexible electronics presents promising future perspectives, driven by advancements in materials, fabrication techniques, and device integration. Ongoing research endeavors aim to refine IDE materials and design to enhance performance parameters such as durability, sensitivity, and conductivity, enabling IDE-based devices to find increased utility across industries like biomedical devices, energy harvesting, and sensors. With IDEs anticipated to be integral to the integration of printed and flexible electronics with emerging technologies like wearable technology, smart fabrics, and the Internet of Things (IoT), the development of small, lightweight, highly functional electronics that seamlessly blend into everyday objects and environments is expected. Additionally, the adaptability of IDE-based devices will spur exploration into new applications in fields like healthcare, environmental monitoring, and human–machine interfaces, with IDE-based sensors poised to enable real-time monitoring of physiological parameters, environmental pollutant detection, and control of robotic systems or prosthetic limbs. Further advancements in additive manufacturing processes such as screen printing, inkjet printing, and 3D printing will make IDE-based device production more scalable and affordable, promoting innovation and accelerating commercialization. Integration of IDEs with flexible substrates like polymers and elastomers will lead to the development of stretchable and conformable electronics, including flexible displays, electronic skins with improved tactile sensing capabilities, and wearable electronics that adapt naturally to the body’s shape. Moreover, the emphasis on sustainability will drive research towards developing IDE-based devices with eco-friendly and biodegradable components, preserving high performance while minimizing environmental impact from materials procurement to end-of-life disposal.

The experimental results from previous literature revealed that IDEs-based sensors exhibit a small capacitance, an aspect that leads to high output voltage in flexible and PE. Several publications included in the study also praise IDEs for being portable, flexible, and durable in nature, making them highly appropriate for use in large-scale and industrial PE production. These advantages demonstrate the vital role played by IDE-based sensors in flexible and PE. The potential of printed IDE technology is immeasurable as many industries, including engineering, medicine, computing, and others, continue to adopt its application. Given the importance of IDE structures, more effort should be dedicated to fully exploring these concepts.

## Figures and Tables

**Figure 1 sensors-24-02717-f001:**
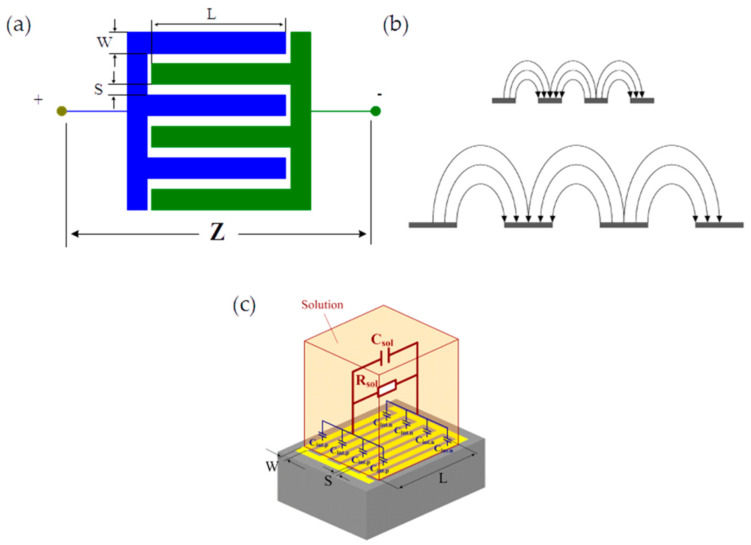
Configuration of IDE structure wherein (**a**) geometric parameters of interdigitated sensors; (**b**) electric current displacement between electrodes; and (**c**) electrical model of an interdigitated sensor and sample (an ionic solution) (Source: [3]).

**Figure 2 sensors-24-02717-f002:**
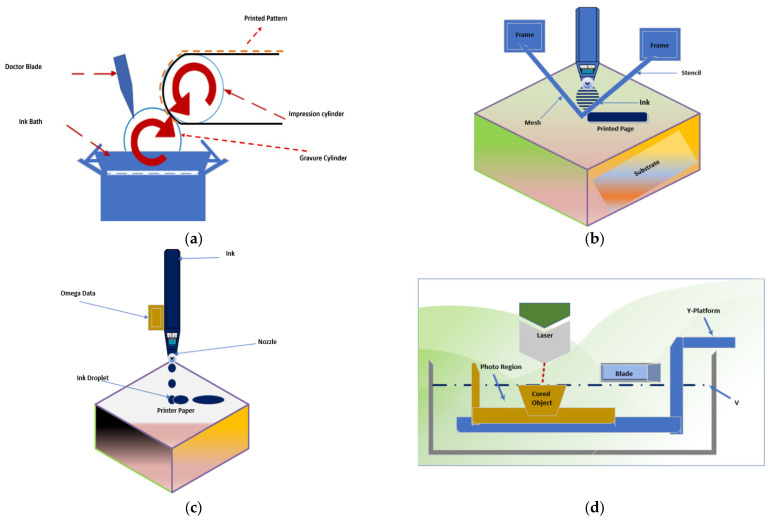
Technology schematics of the main (PE) techniques: (**a**) gravure printing (schemed modified from [31]; (**b**) screen printing; (**c**) inkjet printing; (**d**) 3D printing, (**e**) flexographic printing (schemed modified from [31] and (**f**) laser scribbling (reprint with permission from reference [32]).

**Figure 3 sensors-24-02717-f003:**
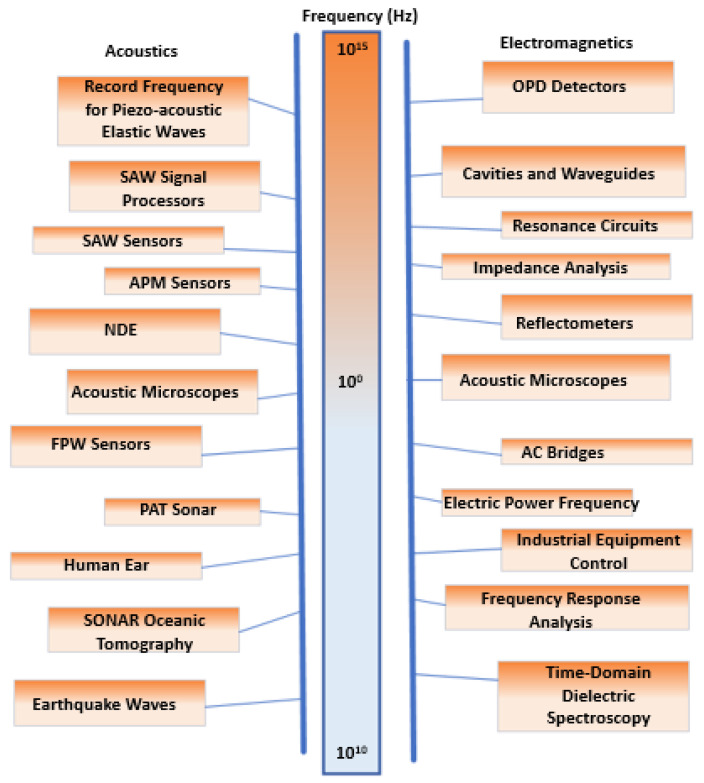
Frequency spectrum for acoustic and electromagnetic digital sensors. OPD: organic photo-diodes, SAW: surface acoustic wave, APM: acoustic plate mode, NDE: node detection emitter, AC: alternating current, FPW: flexural plate-wave, SONAR: sound navigation and ranging.

**Figure 4 sensors-24-02717-f004:**
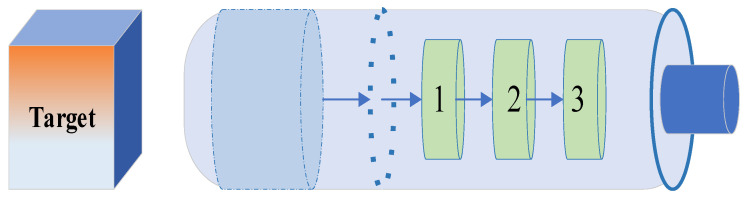
Design of capacitive devices (1—oscillator, 2—trigger circuit, and 3—output switching device).

**Figure 5 sensors-24-02717-f005:**
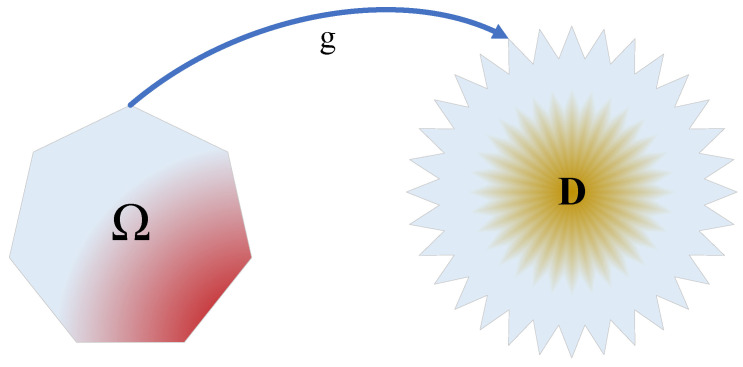
Mapping to the unit disk.

**Figure 6 sensors-24-02717-f006:**
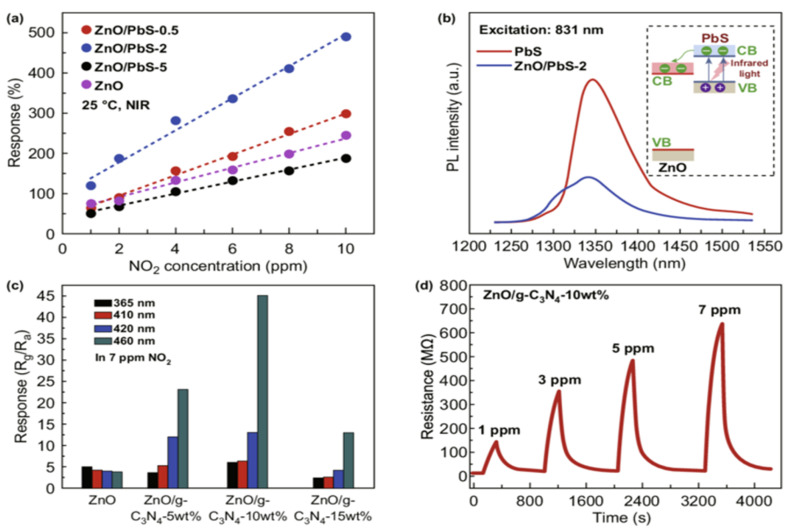
(**a**) Responses of the sensors to 1–10 ppm of NO_2_ at room temperature measured under near-infrared light using ZnO/PbS nanocomposites with varying PbS loading. (**b**) Photoluminescence spectra of ZnO/PbS-2 and PbS excited at 831 nm. Reproduced with permission. (**c**) Reactions of ZnO/g-C_3_N_4_ composites to 7 ppm NO_2_ under varied light illumination wavelengths with varying g-C_3_N_4_ content. (**d**) Dynamic resistance curves for ZnO/g-C_3_N_4_-10 weight percent to 1%#x2013;6 ppm NO_2_ at room temperature under 460 nm light irradiation. Licensed reproduction (reprint with permission from reference [131]).

**Figure 7 sensors-24-02717-f007:**
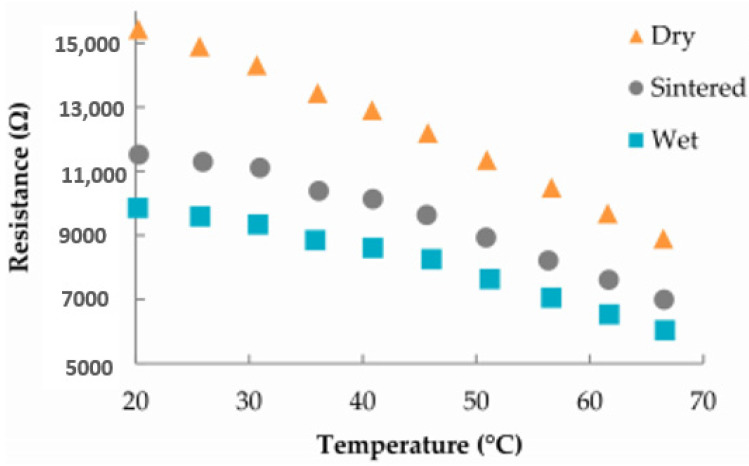
Resistance vs. temperature at 55%RH for electrodes with a width of 150 μm and spacing of 200 μm was studied, showing the temperature-dependent behavior of the electrode material [97].

**Figure 8 sensors-24-02717-f008:**
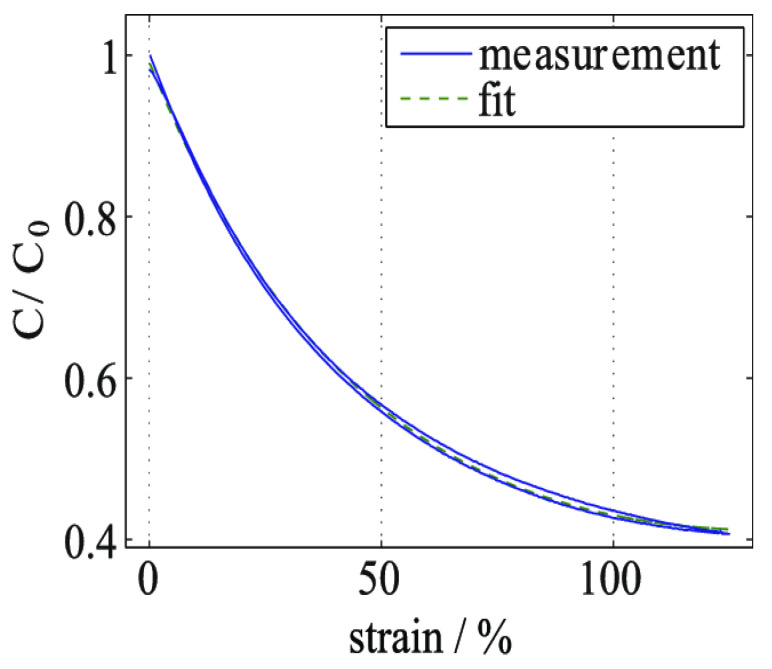
Capacitance strain measurement for the capacitive strain gauge shown in Figure 8 undergoing a strain of 125%. The fit was calculated using the equation = + ϵ + C × c c [135].

**Table 1 sensors-24-02717-t001:** Comparison between different types of printing.

Parameters	Inkjet Printing [51]	Screen Printing [52]	Gravure [52]	Flexographic [53]	Laser Scribing [54]	3D Printing [53]
Required Solution % *w*/*v*	0.002–0.10	0.500–5	0.01–1.1	0.01–0.5	0.01–1.1	0.01–5
Viscosity (Pa·S)	0.01–0.5	3–30	0.02–12	0.17–8	0.02–2	0.01–0.5
Thickness (μm)	0.02–5	0.6–100	8–100	5–180	0.500–1	0.01–1
Printing Speed (m/min)	No	Yes	Yes	Yes	No	No
Material Wastage	No	No	No	No	Yes	No
Resolution	High	Medium–High	Medium	Medium	High	Medium
Accuracy	High	Medium	Medium	Medium	Medium–Low	Medium

**Table 2 sensors-24-02717-t002:** Comparison of various magnetic field sensors [80].

Geometrical Mean Distance (GMR)	H Range (T)	Sensitivity (V/T)
GMR	10^−12^ to 10^−2^	120
Hall	10^−6^ to 10^2^	0.65
Squid	10^−14^ to 10^−6^	1014

**Table 3 sensors-24-02717-t003:** Electric values used in IDE optimization [120].

Material	Relative Permittivity	Resistivity (Ω·m)
Air	1.0005	3.007
Lead zirconate titanate (PZT) [64,92]	X-900 Y-900 Z-1100	10
Electrode	0	10–14
Substrate	4	107

**Table 4 sensors-24-02717-t004:** Comparison of capacitive, resistive, and piezoelectric sensors.

Sensing Technologies	Capacitive	Resistive	Piezoelectric
Maximum Range	Good	Excellent	Fair
Sensitivity	Excellent	Poor	Fair
Minimum Element Size	Fair	Excellent	Poor
Repeatability	Excellent	Fair	Poor
Temperature Stability	Excellent	Fair	Poor
Design Flexibility	Excellent	Fair	Fair

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
