# Peer review of "The Role of Interdigitated Electrodes in Printed and Flexible Electronics"

_sensors, 2024, doi:10.3390/s24092717_

Round 1

Reviewer 1 Report

Comments and Suggestions for Authors

Dear authors, consider the following (in the order of meeting first in the text):

- you mention IDT for piezoelectric sensors - note and clarify that it can be applied only for specific orientation of the crystal, obtaining stronger piezoelectric coefficient in lateral direction (d31 instead of d33).

- surface acoustic wave-based sensors are not discussed as one of the major representatives of the IDT electrodes in their design. Recent progress in the topologies of the SAW-based sensor must be made in the context of the sensor performance due to the IDT design.

- regarding the use of references 12 and 13 in the text related to "recent estimates" should be reconsidered, because ref. 12 is from 2011 and this prognosis has been changed for 13 years. Reference 13 is not sufficiently relevant to the claim that you want to support with this reference.

- you involve screen printing, which deals with thicker films, not very suitable for the flexible electronics due to the impact of the mechanical stress in the thicker film compared to the thinner ones. If you want to keep this section, it is recommended to investigate more about the design and technology of screen printed thick film sensors with details about the mechanical stability of the coatings.

- different methods result in different resolutions of the patterns. Discuss the role of the resolution of IDT on the sensor performance.

- in table 1 add two new columns for the resolution and accuracy of reproducing the desired size of IDT for the different fabrication methods, as this is directly related to the quality of the sensor characteristics.

- mechanical stability must be discussed in a separate section. This is a crucial moment for flexible electronics.

- Section 9 for the energy devices is incomplete, as piezoelectric and triboelectric harvesters use a lot of the design of ITD for specific conditions of charge carrier collection. They are typically wearable, so the context of the flexible electronics is very strong. I recommend more attention be paid to studies devoted to printed polymeric flexible energy harvesting elements (in particular, the impact of the electrode materials and patterns, as related to the durability of the device for IDT design).

- from this conclusion it is not clear what the future perspectives are.

Author Response

We would like to express our sincere gratitude for taking the time to review our manuscript titled “The Role of Interdigitated Electrodes in Printed and Flexible Electronics” and for providing valuable feedback. Your insightful comments and constructive criticism have been instrumental in shaping the final version of our work. In this response letter, we address each of your comments and suggestions in detail, outlining the revisions we have made accordingly, and we hope that the revised manuscript reflects our dedication to addressing the concerns raised.

Reviewer 2 Report

Comments and Suggestions for Authors

sensors-2891744

Title: The Role of Interdigitated Electrodes in Printed and Flexible Electronics

Indeed, the manuscript is well-written and easy to follow. Some points need to be known.

-the quality of all the figures should be improved, especially Figures 6, 7, and 8.

-variables used in equations should be adequately explained e.g. in equation1, Cair, Esub, Csub etc.)

-What do you mean by equation2 ? C=NLC. How is it possible?

- All the equations should be reviewed and rewritten with proper references.

-Section 6 (Modeling of transducers based on interdigitated electrodes) should be rewritten. Proper refernces should be used while writing this section, e.g.

1.          Igreja, R.; Dias, C.J. Analytical evaluation of the interdigital electrodes capacitance for a multilayered structure. Sens. Actuators A Phys. 2004, 112, 291–301.

2.          Igreja; Rui; Dias, C.J. Extension to the analytical model of the interdigital electrodes capacitance for a multilayered structure. Sens. Actuators A Phys. 2011, 172, 392–399.

3.          Khan, A.U.; Khan, M.E.; Hasan, M.; Alhazmi, W.; Zakri, W. Analytical Evaluation of a Coplanar Interdigitated Sensor Capacitance for 1-n-1 Multilayered Structure. IEEE Trans. Instrum. Meas. 2023, 72, 9502410.

4.          Blume, S. O. P., Ben-Mrad, R., & Sullivan, P. E. (2015). Modelling the capacitance of multi-layer conductorfacing interdigitated electrode structures. Sensors and Actuators B: Chemical, 213, 423–433.

doi:10.1016/j.snb.2015.02.088

-Section 7.2 should not be a subsection. 

-Section 7.3: Are Selectivity, sensitivity etc., chemical properties?

-The novelty of the work should be clearly highlighted (in the abstract and the conclusions).

Comments on the Quality of English Language

Moderate editing of English language required

Author Response

(The authors gave the same response as above.)

Reviewer 3 Report

Comments and Suggestions for Authors

The submitted manuscript (sensors-2891744) seams as a 'good review' about an interesting technological class of the recent IDE-based sensors in printed and flexible electronics.  However, the theoretical  and the experimental support (apart Page15) on electrical/electrochemical sensors' properties (and synergies) is weak, or even absent. Such a support might originate from the commercial sensors' specifications. Since, the electrodes' multiplication (in IDE-sensors) does not increase very much the cost, then say, why some (more reliable) sensors are based on more than two electrodes ?  
Improve the conclusions.
Also, there are some (13) minor notes/corrections:
1. Page05: Typing/syntax error. Remake 'materials.it' as 'materials. It',or 'materials; it', in: "has been widely used to produce a variety of printed materials.it is well-suited for printing short runs".

2. Page07: Typing error. Remake 'electronics.3D' as 'electronics. 3D', in: "industries, including healthcare, environmental monitoring, and consumer electronics.3D printing is".

3. Page08: Typing error. Remake 'industry' as 'industry.', in: "its unique strengths and weaknesses, making it suitable for specific applications and industry".

4. Page09: Typing error. Remake 'by (PE)' as 'by PE', in: "between flexible electronics and PE and highlights the potential offered by (PE) in creating flexible".

5. Page09: Typing error. Remake 'devices.PE' as 'devices. PE', in: "devices.PE offers scalability and cost-effectiveness, enabling the mass production of flexible".

6. Page11: Error in Figure 3. Remake exponents in: "Figure 3. Frequency spectrum for acoustic and electromagnetic digital sensors. OPD: organic photo".

7. Page12: Typing error. Remake 'measurements[74].Recent' as 'measurements[74]. Recent', in: "measurements[74].Recent research in the field of acoustic sensors has explored the integration of".

8. Page17: Typing error. Remake 'dialectical' as 'dielectical', in: "in detecting the dialectical properties of a sample. Common properties of these sensors include real-".

9. Page18: Typing error. Delete 'of temperature ', in: "the parallel resistance with respect to the frequency of temperature is due to the leakage of current".

10. Page21: Typing error. Remake 'tempeature' as 'temperature', in: "as well as ploting of resistance against tempeature. It will also explore the strain, the impact of strain".

11. Page22: Typing error. Remake 'thickest' as 'thickness', in: "sample set with the lowest thickest. The field of IDE-based sensors is highly dependent on the".

12. Page22: Typing error. Remake 'electrostatic ally' as 'electrostatic ally', in: "Being an alternative to batteries, super capacitors mainly store energy electrostatic ally. Because".

13. Page24: Typing error. Remake 'PE s' as 'PEs', in: "highly appropriate for use in large-scale and industrial production of PE s. These advantages".

Author Response

(The authors gave the same response as above.)

Round 2

Reviewer 1 Report

Comments and Suggestions for Authors

You added a lot of new relevant information, which is, however, not supported with sufficient references. There were statements "floating" without citation support everywhere in the newly added section. You have to insert more references (relevant and new) and check for the position of the fig. 6 which is shifted currently.

Author Response

(The authors gave the same response as above.)

Reviewer 2 Report

Comments and Suggestions for Authors

sensors-2891744

·         The Role of Interdigitated Electrodes in Printed and Flexible Electronics

·         Figure 1: s, w, l should be labeled with arrow lines.

·         Table 1: References should be added for all printing methods.

·         Equation 1 and Equation 2 are the same. Please correct. Also, equations are not written properly. For example, in equations 3 and 4, g(z), p (x,y), q (x,y), ζ etc, are not defined.

·         Please explain Figure 5.

·         Below Equation 9: “C represents the permittivity of the vacuum…..”. How is it possible?

·         Page 18: “r is the relative dielectric constant of the dielectric…..” Please correct.

·         There are lots of errors in the manuscript. Equations are repeatedly written with a wrong interpretation. Please rewrite the manuscript carefully.

·         The review report should be re-written properly. Changes should be mentioned in the review report.

Comments on the Quality of English Language

Minor editing of English language required

Author Response

(The authors gave the same response as above.)
